# CYP2C19 and CYP2D6 Genotypes and Metabolizer Status Distribution in a Bulgarian Psychiatric Cohort

**DOI:** 10.3390/jpm12071187

**Published:** 2022-07-21

**Authors:** Hristo Y. Ivanov, Denitsa Grigorova, Volker M. Lauschke, Branimir Velinov, Kaloyan Stoychev, Gergana Kyosovska, Peter Shopov

**Affiliations:** 1Department of Pediatrics and Medical Genetics, Medical University-Plovdiv, 4002 Plovdiv, Bulgaria; 2Department of Medical Genetics, University Hospital St. George-Plovdiv, 4002 Plovdiv, Bulgaria; 3Faculty of Mathematics and Informatics, Sofia University, 5 James Bourchier Blvd., 1164 Sofia, Bulgaria; dgrigorova@fmi.uni-sofia.bg (D.G.); pdshopov@gmail.com (P.S.); 4Big Data for Smart Society Institute, Sofia University, 125 Tsarigradsko Shosse, Bl. 2, 1113 Sofia, Bulgaria; 5Dr. Margarete Fischer-Bosch Institute of Clinical Pharmacology, 70376 Stuttgart, Germany; volker.lauschke@ki.se; 6Department of Physiology and Pharmacology, Karolinska Institutet, 171 77 Stockholm, Sweden; 7University of Tuebingen, Geschwister-Scholl-Platz, 72074 Tübingen, Germany; 8Institute for Medical Research (IMR), 1202 Sofia, Bulgaria; branimir.velinov@pdc-eu.com (B.V.); gergana.kyosovska@pdc-eu.com (G.K.); 9Department of Psychiatry and Medical Psychology, Faculty of Public Health, Medical University-Pleven, 5800 Pleven, Bulgaria; kaloyan_stoichev@abv.bg; 10Department of Psychiatry, University Hospital “Dr. Gerogi Stranski”, 5800 Pleven, Bulgaria; 11Bulgarian Association for Personalised Medicine (BAPEMED), 1202 Sofia, Bulgaria; 12R PGx Package Team, 1000 Sofia, Bulgaria

**Keywords:** pharmacogenetics, pharmacokinetics, CYP2D6, CYP2C19, population genetics, cytochrome P450, precision public health

## Abstract

CYP2D6 and CYP2C19 are enzymes of essential significance for the pharmacokinetics of a multitude of commonly used antidepressants, antipsychotics, antiemetics, β-blockers, opioids, antiestrogen, antacids, etc. Polymorphisms in the respective genes are well established as resulting in functional differences, which in turn can impact safety and efficacy. Importantly, the prevalence of genetic *CYP2D6* and *CYP2C19* variability differs drastically between populations. Drawing on the limited information concerning genotype frequencies in Bulgaria, we here analyzed 742 Bulgarian psychiatric patients predominantly diagnosed with depression and/or anxiety. Specifically, we analyzed frequencies of *CYPC19*2*, **4* and **17*, as well as of *CYP2D6*2*, **3*, **4*, **5*, **6*, **10* and **41*. In total, 571 out of 742 patients (77%) carried at least one variant which impacts metabolizer status. Overall, 48.6% of the studied individuals were classified as non-normal metabolizers of CYP2D6 with most exhibiting reduced function (38.2% intermediate metabolizers and 6.6% poor metabolizers). In contrast, for CYP2C19, the majority of non-normal metabolizers showed increased functionality (28.9% rapid and 5.5% ultrarapid metabolizers), while reduced activity metabolizer status accounted for 25.6% (23.8% intermediate and 1.8% poor metabolizers). These results provide an important resource to assess the genetically encoded functional variability of CYP2D6 and CYP2C19 which may have significant implications for precision medicine in Bulgarian psychiatry practice.

## 1. Introduction

Inter-individual differences in drug response constitute an important issue impacting both patient care and drug development. Besides demographic, dietary and environmental influence, genetic polymorphisms are among the main factors responsible for patient-specific differences in drug response or safety. Consideration of the genetic variability in pharmacogenes aims to guide drug selection and dosing in order to provide individualized treatment decisions that reduce morbidity and increase efficacy. As of June 2022, 391 drug labels contain pharmacogenetic information by the U.S. Food & Drug Administration (FDA), and the European Medicines Agency (EMA) has approved 145. This information could guide clinical decision making.

Outside of oncology, the majority of clinically relevant pharmacogenomic biomarkers reside in genes involved in drug pharmacokinetics. Genes encoding cytochrome P450 (CYP) enzymes, specifically *CYP2D6* and *CYP2C19*, contain the highest numbers of actionable associations [1]. Both genes are highly polymorphic and their respective gene products metabolize around 20% and 7% of all clinically used drugs [2]. Particularly in psychiatry, a multitude of commonly used drugs are metabolized by CYP2D6 and CYP2C19, including the tricyclic antidepressants amitriptyline, imipramine and doxepin, selective serotonin reuptake inhibitors, such as paroxetine, fluvoxamine, citalopram and sertraline, as well as various typical and atypical antipsychotics, such as haloperidol, risperidone and aripiprazole. Genetic variations in *CYP* genes can be used to infer functional phenotypes, which are typically categorized: for CYP2C19, into ultrarapid metabolizers (UM), rapid metabolizers (RM), normal metabolizers (NM), intermediate metabolizers (IM) and poor metabolizers (PM); and for CYP2D6, into UM, NM, IM, PM. Considering the high degree of polymorphism of *CYP2D6* (129 known allelic variants according to CPIC allele definition table) and *CYP2C19* (35 clinically relevant variants according to CPIC allele definition table) genes, their influence on the metabolism of a large number of drugs and their overall clinical relevance [3], both genes are included in the practical guidelines of several independent institutions, such as CPIC (Clinical Pharmacogenetics Implementation Consortium) and DPWG (Dutch Pharmacogenetics Working Group).

Frequencies of *CYP2C19* and *CYP2D6* polymorphisms have been extensively studied and these investigations revealed pronounced differences in variant prevalence between populations, even within closely adjacent geographic areas [4,5]. In Europe, pharmacogenomic variability of *CYP2C19* and *CYP2D6* have been studied in a multitude of populations [6,7]. However, only few studies evaluated the pharmacogenetic variability in the Bulgarian population [8,9], and none translated frequency data into inferred spectra of phenotypic variability. Very few studies have assessed the frequency of clinically relevant variants in a specific psychiatric patient population referred for testing in the real-world setting with actual clinical intent.

To assess and present the frequencies of clinically significant genetic variants in *CYP2D6* and *CYP2C19* that could affect therapeutic outcomes, we here provide frequencies for nine specific variant alleles of clinical and functional relevance based on genotypes from 742 Bulgarian individuals with psychiatric disorders, independently tested in real-world clinical setting.

## 2. Materials and Methods

### 2.1. Study Cohort

The presented observational retrospective cross-sectional study included a sample of 742 Bulgarian patients, predominantly diagnosed with major depressive disorder (MDD) and generalized anxiety disorder, as well as a smaller percentage affected by other psychiatric disorders such as schizophrenia, bipolar disorder, obsessive compulsive disorder and others. Informed consent was obtained from all subjects involved in the study. All patients were previously genotyped with the pharmacogenetic panel in the period from September 2017 to February 2021. The average age of the patients was 37.1 years (range of 8 to 88 years). In total, 353 of the patients were male and 389 were female.

### 2.2. Genotyping

Genotyping was initiated by clinicians referring patients in a real-world setting with the intent of clinical use of the results. Sample collection was performed using self-sampling buccal-swab kits. DNA was isolated via MagMax 96 DNA Multi Sample magnetic bead technology with the KingFisher Flex Purification System (Thermo Fisher Scientific, Carlsbad, CA, USA). The genotyping was performed by Genomind Ltd. using standard and custom TaqMan reagents (qPCR) for all variants. *CYP2D6* deletions (*CYP2D6*5*) and duplications were tested by PCR using specific primers.

### 2.3. Statistical Analysis

First, we calculated the proportion of the patients, who are carriers of at least one variant which, if present, changes the phenotype metabolizer status. For this purpose, we used only the variants which unambiguously define a “No functional allele” or “Increased function allele”.

Chi-square tests of homogeneity were performed for the variants of interest (in gene *CYP2C19*—rs4244285, rs28399504 and rs12248560; in gene *CYP2D6*—rs16947, rs35742686, rs1065852, rs28371725, rs3892097 and rs5030655), in order to test for identical distributions of the two populations from which the compared samples were obtained. These variants were chosen from the bigger commercial panel of variants tested, mainly based on their capacity to define unambiguously a single star allele with known functional status and having enough data in both studied and 1000 Genomes project samples in order to perform valid tests and calculations. Variants which take part in multiple allele definitions and impose any ambiguity about the specific star allele and functional status or do not have enough data to be analyzed were not included with the following exceptions: (1) variant rs16947 takes part in multiple star allele definitions; however, it is frequently used to define normal function *CYP2D6*2* allele in the absence of other variants; (2) variants rs1065852 and rs28371725 are often used to define *CYP2D6*10* and *CYP2D6*41*, respectively, despite the fact that they are not specific for the corresponding star alleles. However, they are often used by commercial labs and were also included in the recent recommendation for clinical CYP2D6 genotyping allele selection [10], so we also included them in our study as exceptions to the above mentioned. The calculations were performed on R (R Core Team [11]), version 4.0.4 using the base function “chisq.test”. The genotypes distribution of our study sample (742 individuals) was compared to the genotype distribution of the 1000 Genomes sample (503 individuals, phase_3, European ancestry). It should be noted that for the variant rs35742686 we merged the genotypes that are related to slower (intermediate and poor metabolism—A/DEL and DEL/DEL) than the ‘normal’ extensive metabolism in order to produce statistically more reliable results given the small number of people with genotype DEL/DEL in both samples. We have also performed a chi-square test for homogeneity in order to compare the distribution of variants rs12248560 (**17*) and rs4244285 (**2*) in *CYP2C19*, and rs3892097 (**4*) in *CYP2D6* in our study sample to the same variants in Bulgarian subjects without psychiatric disorders, as previously reported by Pendicheva [9].

## 3. Results

We here present the calculated frequencies of the variants, respective star alleles, diplotypes and metabolizer status for *CYP2C19* and *CYP2D6* genes. The results are described in Table 1, Table 2, Table 3, Table 4, Table 5 and Table 6. Translation to star alleles is performed according to CPIC allele definition tables with the use of genetic variants, which unambiguously define a star allele with known functional status:⮚*CYP2C19* variants: rs12248560 (**17, *4*), rs4244285 (**2*), rs28399504 (**4*)⮚*CYP2D6* variants: rs3892097 (**4*), rs35742686 (**3*), rs5030655 (**6*), rs16947 (**2*) (normal function if no variants, part of another star allele definitions are present) and copy number variations (**5*, **1xN*, **2xN*).

Calculations for *CYP2D6* are based on 560 cases, and for *CYP2C19* they are based on the cohort of 732 cases, after exclusion of all cases for which genotype data is not specific enough to assign a star allele unambiguously.

### 3.1. Genetic Variability Distribution of CYP2C19 in a Bulgarian Psychiatric Cohort

Overall, we found that the gain-of-function allele *CYP2C19*17* is most prevalent with frequencies of 22.84%, whereas the loss-of-function alleles **2* and **4* were found with frequencies of 13.07% and 0.61%, respectively (Table 1, column MAF (%); MAF—Minor allele frequency). For diplotypes, *1/*1, *1/*17 and *1/*2 were most common, accounting for 40%, 29% and 16.5% of all patients, respectively (Table 2).

### 3.2. Genetic Variability Distribution of CYP2D6 in Bulgaria

We found that the variant rs3892097 defining the loss-of-function variant *CYP2D6*4* and variant rs28371725 commonly used as a tag SNP, although not specific, for the reduced activity allele *CYP2D6*41* are the most common *CYP2D6* variations in the study sample with frequencies of 19.20% and 10.38%, respectively (Table 3). In contrast, CYP2D6*3 and *6 are rare with minor allele frequencies (MAF) < 1%.

Additionally, we identified the frequencies of *CYP2D6*5* (*CYP2D6* gene deletion—DEL) and of functional gene duplications (*CYP2D6*1 × N* or *CYP2D6*2 × N*—DUP) in the study cohort (Table 4).

The diplotype composition of *CYP2D6* is substantially more complex than for *CYP2C19* (Table 5). Only 18% of patients carried the *1/*1 diplotype with a further 25.7% and 7.7% carrying the functionally normal *1/*2 and *2/*2 diplotypes. Further relatively common diplotypes were *1/*4, *2/*4, *4/*4, *1/*5 and *2/*5 which were found in 18.8%, 13%, 5.9%, 2.1% and 1.4% of individuals.

### 3.3. Translation of Genotypes into Inferred Functional Consequences

Next, we integrated the obtained genetic profiles to calculate the distribution of metabolizer phenotypes in the study sample (Table 6). To this end, we used the genotype-to-phenotype consensus translations from CPIC. First, we calculated the proportion of the patients, who are carriers of at least one variant that changes the phenotype metabolizer status and found that 76.95% of individuals carried at least one such variation with 95% confidence interval (CI) for the proportion: (73.72%, 79.90%).

For *CYP2C19*, 23.8% and 1.8% of individuals were classified as intermediate or poor metabolizers, primarily driven by the high prevalence of the loss-of-function allele *CYP2C19*2*. Furthermore, 28.9% and 5.5% of patients were predicted to have increased CYP2C19 activity due to heterozygosity or homozygosity of *CYP2C19*17*, respectively.

For *CYP2D6*, 51.4% of patients were estimated to be phenotypically normal based on genetic data, while the vast majority of non-normal phenotypes were predicted to be in the reduced function spectrum with 38.2% and 6.6% of individuals being classified as intermediate and poor metabolizers, respectively. In contrast, only 3.8% of individuals harbor diplotypes that translate into increased function phenotypes.

Combined, these results demonstrate that a large fraction of the studied Bulgarian psychiatric cohort carries clinically actionable alleles in *CYP2C19* and *CYP2D6*. Consequently, consideration of genetic data into selection and dosing of substrates of the respective enzymes promises to improve treatment outcomes and public health.

The test-statistics (chi-square), degrees of freedom (df) and the *p*-values of the tests are given in Table 1 and Table 3 for each gene. For the diplotype distributions of variant rs12248560 (*CYP2C19*17*) it was observed that two samples (study sample vs. Pendicheva’s sample) come from populations with different distributions (*p*-value = 0.04). For the diplotype distributions of variant rs5030655 (*CYP2D6*6*) we found that two samples (study sample vs. 1000 Genomes project) come from populations with statistically different distributions (*p*-value = 0.017). We also have statistically significant difference for the diplotypes of variant rs35742686 (*CYP2D6*3*) (*p*-value = 0.044) for which we tested the reference homozygous diplotype against combined heterozygous and alternate homozygous genotypes. Despite the statistically different distributions found for the variant rs35742686 (*CYP2D6*3*) (study sample compared to 1000 Genomes project), the *p*-value (*p*-value = 0.044) is close to the significance level of 5% and the results should be interpreted with caution. For the variant *CYP2D6*6* for which we found statistically significant difference in distribution when comparing the Bulgarian psychiatric sample and 1000 Genomes project sample, in the Bulgarian sample we observe lower proportion of the genotype T/DEL (1.62% versus 3.98%). Analogous is the situation with the variant *CYP2D6*3* where the proportion of combined genotypes DEL/DEL and A/DEL is 1.35% in the Bulgarian psychiatric sample while in 1000 Genomes project sample it is 3.18%. Regarding the difference in the genotype’s distribution found between the studied Bulgarian psychiatric cohort and Pendicheva’s control sample for variant *CYP2C19*17*, again the direction is that the carriers of the alternative variant (genotypes T/T and C/T) are with lower proportion in the studied Bulgarian psychiatric cohort (40.3% versus 51.41%). As there were no pre-defined hypotheses about the direction of the differences, it was decided to not perform any post hoc statistical tests. Additional studies are needed to confirm our findings.

Overall, the presented proportion in percentage of the variants in the study cohort might be used to supplement the hitherto missing information on the frequency of these variants in the Bulgarian population ([7] Petrovic et al., 2020), with the remark that the study sample is patient specific.

## 4. Discussion

Pharmacogenetics guidelines and recommendations for *CYP2D6* and *CYP2C19* have been published by various pharmacogenetic expert workgroups, such as the Clinical Pharmacogenetics Implementation Consortium (CPIC) and the Dutch Pharmacogenetics Working Group (DPWG), as well as regulatory agencies, including the U.S. Food & Drug Administration (FDA) [12,13].

Based on specific search, there are few studies related to CYP metabolizer status in psychiatric patient-specific group and most of them are focused on specific populations with sample size of around hundred patients [14,15,16,17,18]. At the same time, the studies about *CYP2C19* and *CYP2D6* clinically relevant genotypes in Bulgarian individuals are very scarce.

In the present cross-sectional observational retrospective study, we demonstrated that via the application of pharmacogenetic tests in the real clinical practice in patients with psychiatric diseases an immense number of clinically significant variants are identified, which could potentially influence the use of medicines and could have significance for optimization of the therapy in a huge number of patients by personalization of treatment regimen and/or dosage. Based on our findings and data from the Bulgarian National Health Insurance Fund (NHIF) for the period January—December 2021 related to number of prescriptions and health insured individuals treated with psychopharmaceuticals (antidepressants and antipsychotics, that are metabolized by CYP2C19—escitalopram, and CYP2D6—paroxetine, aripiprazole, risperidone and haloperidol), we may assume that pharmacogenetic testing would identify approximately 78,316 patients with altered metabolizer status (PM, IM, RM (only for CYP2C19), UM) annually. From these patients, 24,622 actionable recommendations might be applicable, and modifications may be warranted due to potential drug–gene interactions. By mapping the prevalence of *CYP2C19* and *CYP2D6* variations in Bulgarian psychiatric patients, our results can indicate the potential added value for pharmacogenetic testing in such cohorts to optimize treatment in accordance with established guidelines.

Since a significant number of previous studies highlight the relationship between CYP altered metabolizer status (PMs and IMs) and the manifestation of side effects which can increase patient morbidity and reduce drug adherence [19], determining *CYP2C19* and *CYP2D6* metabolizer status might be a powerful instrument to improve treatment efficacy. Additionally, by following pharmacogenetic guidelines, adverse events related to specific CYP metabolizer profile could be minimized resulting in positive effects for the quality of life of psychiatric patients. Moreover, based on pharmacogenetic guidelines, a huge number of prescriptions of potentially affected drugs could be optimized and this could affect favorably public health. Besides its relation to drug tolerance (adverse drug reactions), CYP2C19 polymorphisms are also studied in regard to their potential role in interindividual susceptibility to psychiatric disorders. Sim et al.’s [20] study of 1472 European subjects from the Swedish Twin Registry demonstrated that PMs who lack CYP2C19 activity display lower levels of depressive symptoms than NMs. Jukic et al. [21] demonstrated that the absence of CYP2C19 correlated with a lower prevalence of major depressive disorder and depression severity. Furthermore, genetically determined high CYP2C19 enzymatic activity was associated with higher suicidal tendency in depressed suicide attempters.

Although our study on Bulgarian psychiatric patients did not investigate the relationship between *CYP2D6* and *CYP2C19* SNPs and the therapeutic response, future studies with similar or even bigger sample sizes could confirm or rule out the predictive value of cytochrome P450 genetic polymorphisms in terms of pharmacotherapeutic efficacy.

## 5. Study Limitations

Our study had several limitations. First, the study cohort is patient specific which might reduce generalizability. Besides this fact, it is valuable to have patient-specific information, since it represents the real-world practical significance of pharmacogenetic implementation. The impossibility of generalizing the observed results is also deepened by the fact that the studied individuals are mainly from two large cities in Bulgaria (Sofia and Plovdiv) and do not represent a country-wide sample. Additionally, all patients included in the study financed the performed pharmacogenetic test out-of-pocket (OOP). Both life in a large city and financial ability to pay for the pharmacogenetic test are positively correspond with the access to health, and we may assume that people lacking healthcare access are underrepresented in the study patient cohort. As all study participants appear to have medication needs, there might be a risk that the study sample is overrepresented by individuals with phenotypes that cause them to metabolize antidepressants and/or antipsychotic medications poorly. On the other hand, this limitation is addressed indirectly by the chi-square testing against the distributions from 1000 Genomes Project, European sub-population, which is expected to be representative for the general population with European ancestry. Finally, besides the demographic information presented in the “Study Cohort” section, the observational study does not include details related to study subjects as well as subsequent follow-ups of patients (not a longitudinal study), and no additional data were collected and no hypotheses could be studied for arising subsequent events.

## 6. Conclusions

Within this current study, we analyzed data on clinically significant genetic variants in *CYP2D6* and *CYP2C19* genes from Bulgarian psychiatric patients who were tested with a pharmacogenetic panel in real-world clinical settings. By doing this, we provided frequencies for nine specific variant alleles of clinical and functional relevance. Based on the observations and the data available, we may assume that pharmacogenetic testing in psychiatry disorders in the Bulgarian population have promising potential given the fact that the field of pharmacogenetics is well-standardized by clinical implementational recommendations of different institutions such as CPIC, DPWG, etc. A significant number of psychopharmaceuticals prescriptions might be influenced and refined based on pharmacogenetic testing of both *CYP2D6* and *CYP2C19* genes. As there are only three of twelve compared distributions which are shown to not be identical between the samples (study psychiatric cohort vs. 1000 Genomes project sample or Pendicheva et al. sample), and moreover, some of their corresponding *p*-values are relatively close to significance (study sample vs. Pendicheva et al. sample—rs12248560, *p*-value = 0.04; study sample vs. 1000 Genomes sample rs35742686, *p*-value = 0.04 and rs5030655, *p*-value = 0.02), we would cautiously assume that the magnitude of these differences is not significant enough, and in fact, we observe a similar level of polymorphism in the two genes between the samples. Keeping in mind that the profile of samples is relatively different in nature (patient-specific vs. general population) and accepting such minor fluctuations as normal, we would generally state that the frequencies of functionally important genetic variants in Bulgarian psychiatric patients correspond to the expected distribution in European individuals.

## Figures and Tables

**Table 1 jpm-12-01187-t001:** Haplotype and diplotype frequencies of *CYP2C19*.

SNP	N	Genotype			Χ^2^	*p*-Value	Df	MAF (%)	
		** *CYP2C19* **						**Study Sample**	**gnomAD**
**rs28399504 (*CYP2C19*4*)**		**AA (%)**	**AG (%)**					**0.61**	**0.34**
Bulgarian psychiatric cohort	742	733 (98.79)	9 (1.21)		2.7	0.1 ^i^	1		
1000 Genomes	503	502 (99.80)	1 (0.20)						
**rs12248560** **(*CYP2C19*17*)**		**CC (%)**	**CT (%)**	**TT (%)**				**22.84**	**N/A**
Bulgarian psychiatric cohort	742	443 (59.70)	259 (34.91)	40 (5.39)	0.72	0.7 ^i^	2		
1000 Genomes	503	300 (59.64)	181 (35.98)	22 (4.37)					
Pendicheva et al., 2017	142	69 (48.59)	65 (45.77)	8 (5.63)	6.4	0.04 ^ii^	2		
**rs4244285 (*CYP2C19*2*)**		**AA (%)**	**GA (%)**	**GG (%)**				**13.07**	**14.73**
Bulgarian psychiatric cohort	742	12 (1.62)	170 (22.91)	560 (75.47)	2.5	0.3 ^i^	2		
1000 Genomes	503	6 (1.19)	134 (26.64)	363 (72.17)					
Pendicheva et al., 2017	142	4 (2.82)	32 (22.54)	106 (74.65)	0.97	0.6 ^ii^	2		

^i^ X2 test for Bulgarian psychiatric cohort (study sample) and 1000 Genomes Project; ^ii^ X2 test for Bulgarian psychiatric cohort (study sample) and Pendicheva’s sample.

**Table 2 jpm-12-01187-t002:** Diplotype distribution for *CYP2C19*.

	*1/*1	*1/*17	*1/*2	*1/*4	*17/*17	*2/*17	*2/*2	*2/*4
**N**	293	212	121	7	40	46	12	1
**%**	40.0	29.0	16.5	0.96	5.5	6.3	1.6	0.14
**Phenotype**	NM	RM	IM	IM	UM	IM	PM	PM

**Table 3 jpm-12-01187-t003:** Haplotype and diplotype frequencies of *CYP2D6*.

SNP	N	Genotype			Χ^2^	*p*-Value	Df	MAF (%)	
		** *CYP2D6* **						**Study Sample**	**gnomAD**
**rs35742686 (*CYP2D6*3*)**		**DEL/DEL; A/DEL (%)**	**A/A (%)**					**0.74**	**0.98**
Bulgarian psychiatric cohort	742	10 (1.35)	732 (98.65)		4.1	0.04 ^i^	1		
1000 Genomes	503	16 (3.18)	487 (96.82)						
**rs1065852**		**CC (%)**	**CT (%)**	**TT (%)**				**20.69**	**21.93**
Bulgarian psychiatric cohort	742	470 (63.34)	237 (31.94)	35 (4.72)	0.34	0.8 ^i^	2		
1000 Genomes	503	325 (64.61)	153 (30.42)	25 (4.97)					
**rs28371725**		**CC (%)**	**CT (%)**	**TT (%)**				**10.38**	**11.50**
Bulgarian psychiatric cohort	742	603 (81.27)	124 (16.71)	15 (2.02)	0.87	0.6 ^i^	2		
1000 Genomes	503	416 (82.70)	80 (15.90)	7 (1.39)					
**rs3892097 (*CYP2D6*4*)**		**GG (%)**	**GA (%)**	**AA (%)**				**19.20**	**19.26**
Bulgarian psychiatric cohort	742	491 (66.17)	217 (29.25)	34 (4.58)	0.22	0.9 ^i^	2		
1000 Genomes	503	339 (67.40)	141 (28.03)	23 (4.57)					
Pendicheva et al., 2017	59	33 (55.93)	23 (38.98)	3 (5.09)	2.6	0.3 ^ii^	2		
**rs5030655** **(*CYP2D6*6*)**		**T/DEL (%)**	**T/T (%)**					**0.81**	**0.73**
Bulgarian psychiatric cohort	742	12 (1.62)	730 (98.38)		5.8	0.02 ^i^	1		
1000 Genomes	503	20 (3.98)	483 (96.02)						
**rs16947**		**GG (%)**	**GA (%)**	**AA (%)**				**37.26**	**37.58**
Bulgarian psychiatric cohort	742	296 (39.89)	339 (45.69)	107 (14.42)	5.72	0.06 ^i^	2		
1000 Genomes	503	232 (46.12)	197 (39.17)	74 (14.71)					

^i^ X2 test for Bulgarian psychiatric cohort (study sample) and 1000 Genomes Project; ^ii^ X2 test for Bulgarian psychiatric cohort (study sample) and Pendicheva’s sample.

**Table 4 jpm-12-01187-t004:** Diplotype distribution and allele frequencies for *CYP2D6*5* (*CYP2D6* gene deletion—DEL) and functional gene duplications (*CYP2D6*1 × N* or *CYP2D6*2 × N*—DUP).

DUP	NORM/DEL	NORM/NORM	MAF DUP (%)	MAF DEL (%)
47	25	670	3.17	1.68

**Table 5 jpm-12-01187-t005:** Diplotype distribution for *CYP2D6*.

	*1/*1	*1/*1DUP	*1/*2	*1/*2DUP	*1/*3	*1/*4	*1/*6	*2/*2	*2/*2DUP
**N**	101	4	144	13	4	105	5	43	4
**%**	18.0	0.71	25.7	2.3	0.71	18.8	0.89	7.7	0.71
**Phenotype**	NM	UM	NM	UM	IM	IM	IM	NM	UM
	***2/*3**	***2/*4**	***2/*6**	***3/*5**	***4/*4**	***4/*5**	***4/*6**	***1/*5**	***2/*5**
**N**	3	73	4	1	33	1	2	12	8
**%**	0.54	13.0	0.71	0.18	5.9	0.18	0.36	2.1	1.4
**Phenotype**	IM	IM	IM	PM	PM	PM	PM	IM	IM

**Table 6 jpm-12-01187-t006:** Phenotype distributions for CYP2D6 and CYP2C19.

Gene	Phenotype				
	PM (%)	IM (%)	NM (%)	RM (%)	UM (%)
** *CYP2D6* **	37 (6.6)	214 (38.2)	288 (51.4)	N/A	21 (3.8)
** *CYP2C19* **	13 (1.8)	174 (23.8)	293 (40.0)	212 (28.9)	40 (5.5)

## Data Availability

Restrictions apply to the availability of the data. Data was obtained from Genomind Ltd. and their local partner in Bulgaria—NM Genomix, and could be available upon request via cover letter and follow-up approval from the third party/-ies. Provision and access to data is not guaranteed by the author’s team.

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
