# Peer review of "CYP2C19 and CYP2D6 Genotypes and Metabolizer Status Distribution in a Bulgarian Psychiatric Cohort"

_jpm, 2022, doi:10.3390/jpm12071187_

Round 1

Reviewer 1 Report

This manuscript summarizes the frequency of CYP2D6 and CYP2C19 phenotypes in a convenience sample of Bulgarian patients. It is generally well written, but has substantial limitations. More detail is needed about the eligibility standards and recruitment of study participants. All participants appear to have medication needs. There appears to be a very high risk that the study sample is overrepresented by individuals with phenotypes that cause them to metabolize antidepressants or antipsychotic medications poorly.

Also, the subsection titled, “Prescriptions of psychopharmaceuticals, all and potentially affected by pharmacogenetics” should be removed entirely. Are the authors familiar with the PGx practice recommendations for the medications of focus? How many medication orders were re-orders after patients had tolerated initial dosing fine? Even a specific patient had no prior history of the medication and didn’t have a starting dose adjusted per PGx recommendations, should medication re-orders even be considered in the statistical analysis? The findings are tremendously misleading.

Finally, the Discussion section focused on differences in practice guidelines between major PGx consortia and seems disconnected from the empirical analyses. What are the implications of the authors’ findings? How does this study advance the field?

Minor Notes

Intro: CYP2D6 phenotypes are not typically described as in the introduction.

Reviewer 2 Report

The authors introduced CYP2C19 and CYP2D6 genotypes in Bulgarian population. This kind of study was hot issue for bioinformatics. And it gives us extremely massive result, therefore, we can not find key result easily. So, some part should be revised.

Abstract: Too lengthy! Introduction part is lengthy.

Introduction: The goal of this study should be emphasized. And, possible results or hypothesis should be presented.

Method: The statistical methods for survival analysis should be presented.

Discussion: Compared to massive result data, interesting or novel information was weak. There were many previous studies about these genotypes. Its difference with studies in other population and Clinical value of these data should be presented more.

Reviewer 3 Report

1. Author should in detail mention about the role of CYP2D6 and CYP2C19 in the context to examples and drug metabolism. 

2. Author should also mention how certain polymorphism of CYP2C19 has a role in the intensity of symptoms in depression. 

3. Author should mention the negative side and how many people suffer due to drug side effects and this knowledge of CYP would be able to offer a better and more efficient treatment. 

Round 2

Reviewer 1 Report

The authors have extensively revised the manuscript per feedback, and it is a better paper. It should be better, though. First, the authors should provide more detail on the size of findings. Differences were observed between the frequencies in their sample and 1000 Genomes. In what direction? What was the magnitude of the difference? Second, the limitations address chi-square testing against the distributions of the 1000 Genomes Project. Those chi-square tests showed differences. I was looking for a more thoughtful discussion of the sampling frame may affect results and/or differences from the 1000 Genomes Project. I think the data provide important information, but may have limitations that should be identified, even if they can't be addressed.

Second, I still find the "Examples of psychopharmaceuticals that are part of pharmacogenetic guidelines and 297 relevant data from the Bulgarian National Health Insurance Fund (NHIF)" section to be misleading. The analyses are superficial and overstate the current utility of the data. I recommend addresses the points more briefly in the discussion (e.g., something like, "based on our findings and data from ___, pharmacogenomic testing would identify approximately ___ medication orders where modifications may warranted due to potential drug-gene interactions"). The analyses aren't nearly good enough to present as a research finding, but it is helpful to present the findings about the distributions of phenotypes in a way that demonstrates how many medication orders could be affected.

Author Response

Dear Editor,
thank you and the reviewers for the constructive criticism of our manuscript initially entitled "CYP2C19
and CYP2D6 genotypes and metabolizer status distribution in the Bulgarian population". We have
carefully revised our manuscript according to the minor comments and suggestions made by the
reviewers and attach the updated version of the manuscript and a point-by-point rebuttal.
Sincerely,
Dr. Hristo Ivanov on behalf of all authors

Reviewer 2 Report

The authors revised the manuscript according to the comments. Limitation section should be in last part of Discussion section.

Author Response

(The authors gave the same response as above.)
